# Hybrid Leadership in African Neo-Pentecostalism

**Daniel O. Orogun** 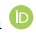

Department of Religion Studies, Faculty of Theology, University of Pretoria, Pretoria 0028, South Africa;
arcorogun2@gmail.com

**Abstract:** Across institutions and professions, leadership philosophy is considered the driver of organisational culture in achieving the overall objectives. Although individuals' leadership cultures may vary, intersections and hybridity are present in many spheres, including that of some African Neo-Pentecostal Leaders (ANPLs). To underscore the hybrid leadership of the ANPLs, qualitative research was conducted, with data collected from 20 participants through one-on-one interviews across Africa. The results revealed the hybridisation of African Neo-Pentecostal leadership styles vis-a-vis African monarchical and religious traditions based on four variables: accountability, ownership and succession plan, healing, and gerontocracy. The results also revealed the benefits and challenges of their hybridity. Subsequently, using Jesus's model of servant leadership to analyse the four variables, the benefits and challenges were critiqued. The analysis identified culture, African spiritual worldview, gerontocracy, and submissive theology as factors influencing such syncretic or hybrid practices. The analysis also delineated the theological, socio-economic, legal, and transgenerational implications of such hybrid leadership. This article concludes with cautionary remarks regarding boundaries, servant leadership, and morality.

**Keywords:** African tradition; gerontocracy; accountability; succession plan; Neo-Pentecostalism; hybridity; syncretism

## 1. Introduction

This article poses a few questions that guide its presentation: Are there syncretic leadership relationships between African traditions and African Neo-Pentecostalism[1], and if so, what is the available evidence? What benefits and challenges can be identified from the available evidence? Is all evidence contrary to Jesus's servant leadership principles? What recommendations can be made to improve the leadership hybridity of African Neo-Pentecostal leaders (ANPLs) vis-a-vis African traditional leadership? To address these questions, this article presents a brief historical and contextual clarification of syncretism in the context of hybridity, evidence of syncretistic practices via qualitative research results obtained from 20 respondents mostly located in Western and Southern Africa, and the benefits, challenges, possible factors, implications, and critiques of the syncretistic relationship. Finally, this article concludes with remarks in the context of Jesus's model of servant leadership, boundaries, and morality.

## 2. A Brief History and Contextual Clarification of Hybridity and Syncretism

In a simple and literal definition, Smith (1974, pp. 1–18) implies that syncretism in anthropology, sociology, and religious studies reflects the assembly and intercourse of different religious traditions. Hartman (1969, p. 7) asserts that it denotes the combination of two or more religions. He cites an example of Hellenistic syncretism, where elements from numerous religions are merged to mutually influence each other. Correspondingly, many authors maintain that syncretism involves combining two or more beliefs into intermingled beliefs and practices (Mullins 2001, p. 809; Pinto 1987, p. 22; Schreiter 2003, pp. 146–47; Droogers 1989, p. 7). However, Hughes (1988, p. 670) believes that syncretic elements

are inherent in all religions, including those with a negative impact on specific religious practices. In other words, syncretism is relative to the definition and usage of the term "religion". Although cultural and religious blends remain the core discussion in this article, the historical perspective will be briefly discussed below.

Ezenweke and Kanu (2012, p. 73) claim that "syncretism" comes from the Greek word *synkretismos*, which originates from the ancient Island of Crete. Noted for their consistent internal crises, the people of Crete often ignored their differences and joined forces to combat external enemies when the need arose. They called this combination of forces *synkretismos*, meaning "to combine". Nyuyki and Van Niekerk (2016, p. 383) add that syncretism has been common throughout history, from the Jewish and Hellenistic worlds to contemporary times. This implies that there is no period without syncretism. Furthermore, syncretism extends beyond religious aspects, as history reveals that even in war, nations and tribes have adopted this principle. This implies that syncretism represents the interaction, intersection, and hybridity of concepts, ideas, and cultures (see Flemming 2007, p. 526). It is little wonder that Hughes (1988, p. 670) asserts that syncretism is relative. Although some older and more conservative authors perceive syncretism from a negative perspective (Nel 2017, p. 4), recent conversations have focused attention on some alternative perspectives. In this regard, Flett (2022) agrees that hybridity offers another way of addressing the questions regarding syncretism. This concept of hybridity is not peculiar to Africa or religions in Africa and cuts across other disciplines and regions of the world (Byron 2021, pp. 425–32; Nel 2017, pp. 4–6; Arroyo 2016, pp. 133–44; Charles 2011, pp. 48–55).

All the above-mentioned definitions and clarifications imply that syncretism can be seen as hybridity in the context of mixing religions, cultures, and other philosophies. Thus, in this article, syncretism is viewed from the perspective of ANPLs' combination of their religious culture with African traditional cultures to provide ministry leadership. In essence, this article will interchangeably use the words syncretism and hybridity; however, the word hybrid was used in the title of this article because it aligns more closely with current conversations around syncretism. The next section examines the relativity of syncretism between African traditions and African Neo-Pentecostal leadership practices.

### 3. Hybridity of African Culture and African Neo-Pentecostal Leadership

Syncretism exists beyond inter- or intra-religious practices. Nyuyki and Van Niekerk (2016, p. 382) opine that Christianity adopted some cultural elements from Judaism and later integrated some Greek and Roman beliefs from the Greco-Roman worlds to sustain its validity. Hartman (1969, p. 8) argues that syncretic phenomena are not limited to religion, but also extend to culture. This is synonymous with the general idea that African Neo-Pentecostalism incorporates some American culture from American Pentecostalism (Orogun and Pillay 2021, pp. 1–18). This blend of culture and religion has been referred to as 'substitution' by Hartman (1969, p. 8), which involves the infiltration of religion by cultural practices or vice visa. For example, the Tanzanian culture of burying the dead outside the family has been adopted and substituted for any church-centred burial rituals (Hartman 1969, p. 140). Hartman (1969, p. 149) also notes the example of an African leader and a Christian leader with a theological degree, who ordered that no religious ceremonies should be held at the time of his death and willed GBP 20,000 to his private sorcerer. This shows how religious leaders can blend their practices with cultural and traditional religions. In the opinion of Johnson (2002, p. 302), this is a syncretic tradition that encourages the hybridisation of religious and cultural elements.

From the perspective of African Neo-Pentecostal Leaders' syncretic culture vis-a-vis African monarchical and traditional religious cultures, it is evident that beyond the Jewish and Greco-Roman cultural influences, African Neo-Pentecostalism has adopted a significant number of practices from African traditions. This is corroborated by Berner (2001, p. 503), who maintains that by regarding "Syncretism on the level of elements", religious groups incorporate elements from different religious systems while concurrently

stressing the boundaries between the systems or even condemning the other system. This is evident in the way some ANPLs condemn African Traditional Religion (ATR) and some monarchical traditions, although traces of these traditions can be found among the ANPLs' practices (see Berner 2001, p. 504). For example, the use of symbolism such as water for feet washing and anointing oil for the covenant of protection are common to both African Neo-Pentecostalism and ATR (see Adogame 2000, pp. 197–99). Additionally, some African monarchical leadership cultures adopted by ANPLs can be seen in their succession plan, gerontocratic leadership, and submissive theology.[2] To support these claims, this article presents evidence from the 20 respondents.

## 4. Evidence of Hybrid Leadership in African Neo-Pentecostalism

One-on-one interviews were carried out with 20 respondents from different parts of Africa. The results show the syncretic relationship between African traditional systems (monarchy and religion) and African Neo-Pentecostal Leaders' (ANPLs) practices. Four questions related to accountability, succession plans and church ownership, gerontocracy, and healing were posed. The interviews covered a wide spectrum of ministry experiences, locations, and affiliations. Due to respondents' requests, some individuals' and Churches' identities were coded to comply with the research ethics. Using a semi-structured question-naire, the respondents were free to respond outside the Neo-Pentecostal scope to provide more references and examples. Table 1 shows the research summary.

**Table 1.** Respondents and their views on the hybrid leadership of African Neo-Pentecostal Leaders.

| S/N | Time | ID Code | Location | Syncretism View | Church Affiliation/Experience |
|-----|------|---------|----------|-----------------|-------------------------------|
| 1 | 90 min | FOPSA | Pretoria, S/Africa | Positive | 12 yrs. as a provincial pastor, LFC. |
| 2 | 100 min | JOPSA | Central African Rep | Positive | 20 yrs. of missionary experience in LFC |
| 3 | 60 min | ULOSA | Maiduguri, Nigeria | Positive | 7 yrs. in Lifaworld Church |
| 4 | 97 min | JEROCK | Abuja, Nigeria | Positive | 21 yrs. of ministry and leadership experience in PHC |
| 5 | 90 min | JAROCK | Abuja, Nigeria | Positive | 20 yrs. as a Pentecostal lay minister |
| 6 | 70 min | CIROCK | Abuja, Nigeria | Positive | 18 yrs. in a pastoral ministry. |
| 7 | 30 min | IMROCK | Kaduna, Nigeria | Positive | 9 yrs. in a PAHOLAC ministry. |
| 8 | 80 min | STAROON | Lagos, Nigeria | Positive | 24 yrs. as a minister and church admin. |
| 9 | 80 min | STARSOK | Lagos, Nigeria | Positive | 19 yrs. as a minister and church admin. |
| 10 | 60 min | SEGSA | Pretoria, S/Africa | Positive | 15 yrs. in ministry |
| 11 | 118 min | CHUDAP | Jos, Nigeria | Positive | 42 yrs. in ministry. PFN/CAN Sec. |
| 12 | 90 min | YAPP | Jos, Nigeria | Positive | 30 yrs. in ministry. P. F.N/CAN National Council member |
| 13 | 95 min | DAPSA | Pretoria, S/Africa | Positive | 30 yrs. in ministry. Chair NPASA. |
| 14 | 45 min | OLAUK | United Kingdom | Positive | Finance consultant. 17 yrs. Pentecostal |
| 15 | 65 min | DRJULIUS | Pretoria, S/Africa | Positive | Pastor and leadership expert (15+ yrs.) |
| 16 | 30 min | DRTHABISO | Free State, S/Africa | Positive | Doctoral researcher |
| 17 | 50 min | MRMUSHI | Pretoria, S/Africa | Positive | Security and leadership expert (35 yrs.) |
| 18 | 45 min | DRJACOB | Free State, S/Africa | Positive | Minister and theology academic (21 yrs.) |
| 19 | 45 min | PJOEL | Zimbabwe | Positive | Pentecostal minister (17 yrs.) |
| 20 | 45 min | PJOSHUA | Nigeria/S/Africa | Positive | Pentecostal minister (16 yrs.) |

### 4.1. Respondents' Views on Accountability

Most respondents believed that some African Neo-Pentecostal Leaders (ANPLs) do not place a high value on being held accountable to their congregants. Ministers FOPSA and JOPSA maintained that accountability in their previous church of primary assignment (LFC) followed a bottom-to-top approach but that the General Overseer (GO) reported to nobody.[3] Ideally, heads of organisations report to a Board of Trustees (BOT); however, in African Neo-Pentecostalism, BOT members report to the GO. Throughout the years that FOPSA and JOPSA were in the LFC, the GO was accountable to no one and was unchallenged. However, ULOSA, another LFC respondent, disagreed. In his words "Church leaders are not designed to be accountable to members except in cases where government policy permits." IMROCK, a currently serving clergy in the PHC (full name withheld), disagreed. He claimed that GOs maintained an 'Alpha and Omega' status in

African Neo-Pentecostalism, making it difficult for members to demand accountability. In his words, 'I have a problem with African Neo-Pentecostal leaders not being accountable'. Likewise, SEGSA agreed that every pastor is accountable first to God and then to his family, church members, government, society, and the universal body of Christ. He believed that the government should regulate churches as they do with other religious entities but argued that the government must never usurp their roles.

Furthermore, in the PHC, JEROCK explained that accountability to members was intentionally avoided using the doctrine of spiritual authority to ensure that the leaders were unquestionable. JAROCK also claimed that there was zero downward accountability in the PHC. Departments give monthly accounts to branch pastors, but the pastors give no account of their stewardship in return. The branch pastors give a financial report to the GO but there is no reciprocal stewardship account. Submissive theology or spiritual authority and 'loyalty and disloyalty' training are used to ensure that devotees are voiceless. CHUDAP maintained that servant leadership, which prioritizes accountability, has been replaced with boss leadership among the ANPLs, hence the prevalence of authoritarianism. YAPP claimed that it is dangerous theology when an ANPL claims, 'I am only accountable to God, if I fail, God will hold me responsible.' YAPP called it a manipulation that promotes one-man business practices in the church and a way of dodging accountability. Conversely, STAROON and STARSOK had different experiences at the DCC (full name withheld), where the GO promotes the policy of accountability. Using the acronym REAL (Righteousness, Excellence, Accountability, and Love), the GO and his leadership team submit themselves to BOT members. The church also reports to the Church Financial Accountability Association (CFAA), a body created by Christian leaders, accountants, lawyers, and auditors to help churches develop a culture of accountability. The final authority in the DCC is vested in the BOT. DAPSA, a Neo-Pentecostal veteran, claimed that most GOs are unquestioned and unaccountable to anyone but God and have successfully given false information about their authority over congregants. OLAUK a financial consultant stated, "I have issues with Neo-Pentecostalism in Africa where church leaders claim to be above scrutiny. Evidence has shown that most of the privileged Pastoral positions are abused".

Furthermore, DRTHABISO asserted that similar to the traditional custom of kings, in South Africa, individuals are not allowed to speak directly to kings and can only pass their messages through the king's designated mouthpieces. Likewise, in most Neo-Pentecostal Churches, leaders are highly revered and one of the ways the culture of reverence is sustained is by surrounding the church leaders with 'Protocol and Security' teams to maintain distance between leaders and members. As such common people have no easy access to their spiritual leaders; a situation which could strengthen leaders' sense of superiority and proclivity for lack of accountability. However, there are exceptions. MRMUSHI reported that post-apartheid monarchical practices in South Africa do not permit tax collection by kings. Rather, kings engage in fundraising activities to carry out projects. They can also sell land to farmers and businesspeople. Giving accounts to the kings' subjects or community is out of the question because the kings are the landowners. In addition, in the traditional practice of "Initiation", the Inyangas (traditional healers) and kings are responsible for the schools of initiations[4]. They collect approximately ZAR 2500 from the families of those who are to be initiated. The amount collected sometimes reaches millions of rand and these funds are not accounted for in the public space. Moreover, Zulus believe that the king never lies and cannot be questioned. In a way, whatever the kings claim regarding the funds is sacrosanct and they do not owe anybody an explanation. This applies to some ANPLs. Some deliverance and healing ministers charge members, as in the case of the Inyangas. Most ANPLs collect tithes and offerings, conduct fundraising for projects, and sell religious items such as anointing oil, water, and handkerchiefs. Regarding the collection amounts, they are accountable to God only and the finances of the church cannot be questioned by its members. Bishops, pastors, and prophets are seen as direct communicators with God and whatever they say is final. Although there may be cabinet members, such as in the case of kings or Board of Trustee members and elders appointed by the church leaders,

they are simply 'rubber stamp appointees' and whatever the leaders say is final. Invariably, there are similarities between the accountability of African traditional leaders and ANPLs in contemporary times.

PJOEL's account of Zimbabwe revealed that leaders are held accountable in established Pentecostal churches as they strive to put systems in place for accountability. Constitutionally, churches in Zimbabwe are expected to have BOTs that require transparency from their leaders. However, in some one-man ministries and equally traditional Apostolic churches where the leaders are seen as semi-divine beings, accountability is out of the question. Referring to a personal experience in South Africa, PJOSHUA cited a case where a female GO threatened to revoke her anointing from her associate when questioned on the poor welfare of associate ministers.

*4.2. Respondents' Views on Church Ownership and Succession Planning*

With 12 years of experience as a provincial head in the LFC (full name withheld), FOPSA alleged that the children of the GO are positioned for transgenerational leadership and that the GO's family runs all the church-owned schools as their personal enterprise. JOPSA alleged that the international headquarters is under the control of the GO's first son. Sadly, most bishops who were ordained ahead of the GO's son now report to him. The second son also controls international missions. Concomitantly, JEROCK claimed that the LFC was built as a family dynasty. Using the constitution of the LFC to substantiate his argument, JARCOK reported that the GO of the LFC claimed he was under God's instruction to copy the succession plan of an American preacher who transferred the church leadership to his son. Furthermore, CHUDAP claimed that some of the businesses that are run by Neo-Pentecostal churches were registered as belonging to individuals or companies with connections to the GOs, allowing church leaders to evade taxes by using the church as an umbrella for their personal businesses. However, ULOSA supported the dynasty or family business model, stating that "Public sentiment is irrelevant as long as the children of the GOs are qualified." IMROCK and SEGSA held the same view as ULOSA, but IMROCK later admitted that selfishness was one of the reasons that GOs retain leadership within their family.

For STAROON, the ministry should not be a family business but rather succession should be based on calling and family members should not be forced into leadership. Referring to Acts 7:27, STAROON noted that Moses was asked, 'Who made you a ruler over us?' Every man of God will answer this question. If a man's father forced him to be a church leader upon the father's passing, people will ask him, 'Who made you a ruler over us, is it your father or God?' STAROON strongly claimed that the case was different in the DCC because the GO's children were not in any church leadership position. Likewise, STARSOK's position was that 'the African monarchical systems among the ANPLs must be challenged. The idea of manipulating the church to ensure that an offspring of the GO becomes the next leader or president of the ministry is inappropriate. He agreed with STAROON that the case is different in the DCC. Regarding the Neo-Pentecostals in general, CHUDAP maintained that 'priesthood is not about inheritance in the New Testament'. The back-door arrangement of transferring leadership to the offspring of GOs is unacceptable. According to YAPP, "The belief that the church must be taken over by the GO's family is completely wrong ... most times the work crashes in the hand of the son or wife after his demise." DAPSA also added that "Most Neo-Pentecostal churches in Nigeria are built around powerful charismatic leaders who raise their children to protect the empire". Referencing late Bishop IDY (real name withheld), OLAUK stated, 'It is ethically incorrect to convert the church to a family asset. Likewise, lifetime headship of the church by a person or one family is unethical.' On a positive note, OLAUK cited the Redeemed Christian church of God (RCCG), where Pa Akindayomi handed over to Enoch Adeboye without any bloodline connections.

DRJULIUS acknowledged that South Africa's social development policies discourage personal ownership of church property. Upon dissolving an organisation, the property

should be transferred to an existing like-minded organisation, but there have been breaches of this policy among the ANPLs, which speaks volumes about their sense of accountability. This entitlement culture is akin to the African monarchical system, where the king is entitled to land ownership. DRJULIUS further noted the syncretic connection between the monarchical system and the Neo-Pentecostal succession plan, where some ANPLs have transferred leadership to their families in the last 40 years, just as in African monarchical leadership. Sometimes when the monarchical culture is ignored, church division ensues. The Christian Missionary Fellowship in Cameroon, founded by Prof Zacharias Tanee Fomum, is a perfect example of this. He did not hand over the church to his son or associates. Upon his demise, through his written will he handed over the church to an unknown missionary without any professorial pedigree, as commanded by God. Although crises and division ensued, the sons and daughters of the founder continued to work with the new, unpopular leader while their mother sided with the founder's former associates who believed that the traditional monarchical system afforded them the right to be appointed as successors. This confirms that not all Pentecostal leaders have a syncretic approach to ownership and succession planning.

For DRTHABISO, kingship in Africa runs genealogically, as it did in the days of David in the Bible. The same is applicable to most African independent and Pentecostal churches in South Africa. According to MRMUSHI, traditional leadership runs in the family. He observed that African traditional and Neo-Pentecostal churches have adopted the same succession plan. For example, the founder of the ZCC in Zimbabwe died and his son took over. Likewise, the leadership of the ZCC in South Africa was passed on to the late founder's son. Currently, the grandson is running the church. Another example is the family battle for leadership in the Shembe Church between Mduduzi and his uncle Vela Shembe in Kwazulu Natal. Among the ANPLs, an example is the current case of the International Pentecostal Holiness Church (IPHC) run by the Modise family. The founder died, handed it over to his son, and now the grandchildren of the founder have been taken to court regarding the church leadership. The case between Prince Simikade and King Misuzulu of the Zulu Kingdom is a similar example. Thus, it can be inferred that just as kings pass on their thrones and lands to their bloodlines, the same is mostly true among African Neo-Pentecostal churches.

PJOEL: In Zimbabwe, church leadership is passed down to the GO's children. In the NLCC in Zimbabwe, the son of the GO is currently the CEO of the church. The son might be the next successor pastorally. PJOSHUA argued for and against this, citing two examples. The first was the RCCG Nigeria, where PA Akindayomi handed over to Enoch Adeboye without a bloodline connection. The second was the case of the Church of God Mission in which Archbishop Idahosa's wife became the successor. In both examples, PJOSHUA argued that the succession plan was about what God said to the GO.

There were strong arguments among the respondents for and against family-based succession. There was also the belief that succession plans in African traditional leadership are genealogical. Although some ANPLs use this type of succession plan, it cannot be said that African monarchs borrowed the idea from ANPLs. The African monarchical custom of bloodline succession existed before the advent and proliferation of Christianity in Africa. Thus, it can be inferred that some ANPLs identified with their African monarchical heritage and incorporated it into church leadership, which was easy because in places such as America, from where most ANPLs find motivation and mentorship, keeping leadership within the family is common practice. Thus, there is a syncretic approach to leadership among the ANPLs in their succession plans. In conclusion, this hybridity may represent a benefit to the ANPLs and congregants. On the other hand, these benefits may not be without their challenges, especially regarding abuses of power. However, it is noteworthy that there are exceptions according to the testimonies of PJOSHUA, STAROCK, STAROON, and DRJULIUS.

*4.3. Respondents' Views on Gerontocracy*

There are opinions for and against gerontocracy in this subsection. DRJULIUS believed that older people had a stronger voice and an upper hand. In the patriarchal African tradition, fathers, men, and older people are accorded higher respect. The older one becomes, the better prepared he is to join community leaders such as chiefs, imams, and 'Lamidos', as in the case of Cameroon. These gerontocratic leaders are sometimes perceived as autocratic authorities in society. As a result, modern churches have adopted these gerontocratic principles from African traditions, which have both positive and negative implications. For example, some gerontocratic principles of leadership represent the virtues of communalism, liberalism, greater security, solidarity, and human dignity. Conversely, today, we have some Christian leaders, especially the ANPLs, who act like traditional chiefs and healers such as sangomas. Today's African church leaders are inspired by the kingship leadership style of traditional systems. MRMUSHI quoted a South African proverb that says, "grey hair is wealth," meaning that the elderly are regarded as wiser because of their life experiences; therefore they have an advantage when it comes to leading and guiding the younger generation. For example, the ZCC, a healing church that requires adherents to provide 24 h service to the church, has more pensioners as clergies and staff because younger people who are busy with money-making ventures are less likely to provide 24 h service to the church. This may be the philosophy behind gerontocracy. However, this is different for African Neo-Pentecostal churches, where young people are on the frontline and the prosperity gospel is at the forefront, making it easier for younger people to embrace. Today, we find so much corruption and crises in Neo-Pentecostal churches because of the popularity of younger leaders who are less committed to service than profit.

In the Zimbabwean context, PJOEL noted the dominance of the older generation in Neo-Pentecostal and African Independent churches. For example, in the AFM (full name withheld), the president, vice president, and Board members belonged to the older generation. Although younger members were being trained in Bible school and given branch churches to lead, the Board and church council were still dominated by the elders. However, in recent times, some churches have been trying to break from this gerontocratic system. This may take much longer than expected because of the traditional belief that the elders are wiser and better equipped for service. This syncretises with the African monarchical system, where the chiefs, elders, and counsellors to the throne are elders, even if the king is young. Lastly, in his 27 years as a Neo-Pentecostal leader, PJOSHUA has seen enough gerontocratic leadership. In the context of Nigeria, he presented a proverb that justifies the gerontocratic culture—"A big Cock does not allow a small Cock to crow." In other words, younger ministers are often silenced and dominated by older leaders. In a situation where a young minister has the elders' support to speak, there is most likely the benefit of Godfatherism. In such cases, the family members, blood relations, wealthy young persons, and ardent loyalists of the GO enjoy such privileges.

*4.4. Respondents' Views on Healing*

DRJULIUS asserted that the church believes in healing and miracles but that extremism in such practices can lead to abuse. Some ANPLs attribute the healing power to them being instruments of God. They assign to themselves the glory component of healing, which is naturally of God. Subsequently, to sustain the glory component, the ANPLs put insurmountable pressure on themselves to live up to expectations. Thus, they do everything possible to display consistent healing powers and miracles in church services. Today, most congregants are desperate for healing miracles, thereby increasing the pressure on the healers. Consequently, some healing and miracle evangelists and prophets now use traditional healing powers. It is then no surprise that they are affiliated with sangomas, black magic, and other diabolical means. Besides using these mediums, some use more sophisticated manipulations to fabricate fake miracles and healings to fill the demand gap and please crowds. Sadly, despite all the syncretic mediums applied, these healing and

miracle Christian leaders cannot live up to expectations because healing comes from God, not charismatic and manipulative methods. In addition, salvation is the ultimate healing.

DRTHABISO opined that South African Christian leaders administered healing and deliverance from traditional African knowledge before accepting Christianity. Upon the intervention of missionaries, they partially embraced some Western religious cultures but did not accept the obliteration of their African healing processes, which promote the sustenance of healing elements, including roots, water, oil, salt, honey, etc. For example, the founder of the ZCC was the son of a chief traditional healer. Although he adopted some of his father's practices such as healing and cleansing using water, he jettisoned the school of initiation. Yes, today's Christian leaders pray in the name of Jesus but concurrently appropriate healing the African way.

Furthermore, MRMUSHI claimed that the APNLs used oil, water, honey, and other items such as towels, handkerchiefs, etc. These practices are similar to African traditional healing processes. Most likely, the ANPLs used such healing methods to discourage their members from looking elsewhere for healing and miracles. Thus, some form of syncretism occurred between African traditions and African Neo-Pentecostalism in healing practices. Lastly, DRJACOB claimed that the ATR believed in the supreme being and spirits of the ancestors. Since many Africans are afraid of witchcraft activities and attacks, when they visit the sangoma (herbalist) they expect to obtain herbs and related elements to cast out evil spirits and witchcraft. This applies to African Neo-Pentecostal churches today. Africans believe that sicknesses and other misfortunes are not only physical but also spiritual. Therefore, both the ATR and ANPLs use elements that speak to the African worldview to provide healing services.

## 5. Discussion: Factors, Implications, and Critique

### 5.1. Factors

From the respondents' data, the factors aiding syncretic leadership among Neo-Pentecostal churches are categorised as cultural issues, African worldview, gerontocratic philosophy, and submissive theology. Each factor is discussed below.

### 5.1.1. Cultural Issues and Accountability

The respondents attested to the similarities between the ANPLs and African traditional leaders' culture of accountability. The question then is, which of these two adopted the practice from the other? It is a custom for African kings not to be accountable to their subjects, just as patriarchy in African families confers respect and authority to father figures. Thus, the kings could not have adopted such a custom from the ANPLs so it must be the other way around. In the final analysis, this hybridity may not represent benefits for the congregants and African society because of possible elements of abuse. Although there are exceptions according to the testimony of STAROON, STARSOK, and DRTHABISO, it is an abuse of power for leaders not to be held accountable to their members and society. Although this may be custom, as in the case of African monarchy, it raises the question of moral responsibility on the part of the ANPLs.

Correspondingly, two respondents associated the challenge of accountability among ANPLs with African culture. Firstly, STAROON opined that in African Neo-Pentecostalism, charisma sets the tone for the hero worship of GOs, which subsequently engenders autocratic decision making. A case in point is the deep-rooted hero worship in the Yoruba monarchical philosophy of 'Kabi-o-osi', a tradition that means that chiefs, fathers, spiritual leaders, and kings are unquestionable[5]. Similarly, the ANPLs are revered as unquestionable superhumans who function as intermediaries between adherents and God and are therefore accountable to no one except God. Such a philosophy creates a gap and encourages a proclivity for manipulation and the extortion of adherents. Thus, it may be challenging to hold leaders with unquestionable superhuman status accountable. When their authority goes unchecked, abuse is inevitable. STARSOK alluded that poverty further strengthens hero worship. In an environment of abject penury, few wealthy individuals are likely to attract

hero worship from the poor majority. With African monarchs and ANPLs surrounding them and their accumulated wealth, they can be easily idolised. Furthermore, Africans believe that kings, chiefs, imams, and pastors have spiritual powers and, therefore, must be revered, honoured, and dreaded. This disposition forms the basis of the non-offensive and non-confrontational approach of adherents, making it difficult to speak truth to power regarding accountability. This then raises questions about the benefits and challenges of syncretic leadership. This article holds that non-accountable leadership culture places a disadvantage on the hybridity of African traditions and African Neo-Pentecostal leadership.

### 5.1.2. African Worldview and Healing in African Neo-Pentecostalism

Africans have a rich spiritual worldview. They believe that they have a spiritual connection to their problems. Hence, they seek spiritual powers. They also think there are intermediaries between God and man who can help to solve their problems (see Mbiti 1976, pp. 75–78). For example, the priests in the African Traditional Religion (ATR) assist people with overcoming unemployment, barrenness, marital issues, and poverty, among others. Africans view these traditional doctors as having spiritual powers and who can provide services at a cost. These services have been adopted by some of the ANPLs. As Van den Torren succinctly stated, "In the African syncretic charismatic churches today, the man of God has replaced the witchdoctor" (Van den Torren 2015, p. 113). Thus, the African spiritual worldview has been integrated by the ANPL, who assert that adherents cannot access God without their mediation (Nyirongo 1997, pp. 51, 54). The question then is why did the ANPLs adopt the African worldview in their healing ministries?

The results revealed three reasons the ANPLs easily adopted syncretic healing processes. Firstly, the ANPLs' African identities suggest an affinity with African spiritual worldviews that subsequently influenced their healing ministrations. Secondly, protecting their churches from interdenominational or inter-religious "sheep stealing" encouraged the ANPLs to administer healing in an African way to avoid members looking elsewhere for healing. Thirdly, the quest to maintain self-glory by meeting the demands for healing, as noted earlier by DRJULIUS, allowed for the mix of divine healing and African traditional healing processes. In the final analysis, this hybridity comes with benefits and challenges. The use of herbs, roots, anointing oil, and related elements is not antithetical to the Christian faith. In other words, the hybridity of healing processes via the combination of herbal therapeutic elements and prayers in the name of Jesus is an appreciable development in African Neo-Pentecostalism. Nevertheless, there are two main challenges of this hybridity. One is the extremism of the hybrid healing process that involves human sacrifice and other questionable practices. There is no provision in the New Testament for such practices in the name of healing in the church. The second is the overdrive of healing and miracles practices, which leads to the commercialisation of the process. Selling handkerchiefs and anointing oil for outrageous prices or charging exorbitant fees for healing consultancy are now popular practices among some ANPLs and represent an abuse of the hybrid healing process in the church. Thus, caution and balance are imperative in the appropriation of hybrid healing ministries among the ANPLs.

### 5.1.3. Gerontocratic Philosophy

The respondents provided valid information that supports the hybrid leadership of ANPLs vis-a-vis African gerontocracy. Preference is often given to age and experience among some African Neo-Pentecostal mega-churches such as the Redeemed Christian Church of God, the Church of God Missions, and the Winners Chapel, all with headquarters in Nigeria. Why did the ANPLs adopt the gerontocratic leadership style? Firstly, the ANPLs' African identities allow them to easily identify with the role of elders in leadership. Secondly, the custom of gerontocracy is an African philosophy. The proverb 'grey hair is wealth', provided earlier by MRMUSHI, represents this. In other words, African gerontocratic philosophy suggests that older people act from life experiences using ancient wisdom. The cornerstone of this idea is that elderly people are more committed to service



while young people are more committed to profit. Perhaps this is the reason the African monarchical Council of Thrones elders are expected to have such a high level of wisdom for their service. Additionally, this philosophy is common in West Africa. In the Yoruba context, it corresponds with the concept of "Omoluabi". Omoluabi can be simply interpreted as one with a virtuous character. Such virtues include courage, hard work, humility, and respect. In African gerontocratic societies, respect for and adherence to elders' decisions are considered the ultimate virtues of an Omoluabi. Subsequently, the concept confers traditional civility upon an Omoluabi and allows the recipient of an Omoluabi's civility disposition (an elder) to enjoy the privilege of supremacy and honour. Thus, most West Africans, especially Nigerians, adopt the posture of an Omoluabi when interacting with elders, kings, chiefs, and spiritual leaders, including the ANPLs. This disposition allows for the easy adoption of gerontocracy by the ANPLs because it facilitates submission to their spiritual authority in the church.

In conclusion, this hybridity represents an appreciable benefit to the ANPLs and the Christian community. It makes leadership less stressful and equal adherence easier within the missional setting. Additionally, adopting the African gerontocratic leadership philosophy allows for quality service and wisdom-aided decision making among the ANPLs. However, one challenge of this hybridity may be the potential for extremism, which can have two outcomes. The first is the propensity for authoritarianism and the second is the tendency to ignore the contributions of the younger generation. Thus, caution becomes imperative for the ANPLs when using gerontocratic leadership.

### 5.1.4. Submissive Theology

The doctrine of spiritual authority is a major tool for exploitation and the abuse of human rights among the ANPLs (Orogun and Pillay 2022, pp. 1–28). This doctrine is synonymous with submissive theology. According to Olaniyan (2009, pp. 88–89), religious leaders secure undeserved advantages when they adopt titles such as Bishop, Reverend, Imam, Prophet, and Alhaji or alhaja (titles obtained after a pilgrimage to Mecca). In Fela's opinion, the conferment of these titles can lead to exploitative practices and submission to religious authorities. According to Fela, the two religions (Christianity and Islam) were founded on submissive theologies) that allow religious leaders to exploit their followers for economic gain, a practice he classified as capitalistic (Olaniyan 2009, p. 89; Moore 2009, p. 47). He called these religious leaders 'profiteering businessmen of God' who indoctrinated congregants to believe that absolute submission to church leadership was a condition for wealth and blessings. With such a submissive theology, followers were psychologically manipulated to hold the view that it was not their place to demand accountability from their religious leaders (Orogun 2020, p. 166). Besides Fela's thoughts, the syncretic relationship between the ANPLs and the African monarchical leadership contributed to the spread of submissive theology in Africa. Africans assumed the posture of submission to kings, chiefs, lamidos, and community leaders, a submissive practice that can confer a status of unquestionability on African traditional and religious leaders. As most respondents opined, the ANPLs are unquestionable because they are seen as being closer to God and are therefore not subject to scrutiny. Lumumba (2015) acknowledged that, consequently, it may be very difficult to curb this belief because when leaders invoke the name of God the congregation is frozen, the people are powerless, and the interrogation of the leadership becomes more difficult. The question then is, what motivates the ANPLs to propagate submissive theology? The ANPLs' theology found common ground in African views such as gerontocratic philosophy and the Kabi-o-osi and Omoluabi concepts. Thus, it can be inferred that ANPLs' submissive theology was not founded solely on biblical theology but also on the hybridity of African culture and religious tenets.

*5.2. Implications*

This article categorises the implications of syncretic leadership among the Neo-Pentecostals as theological, socio-economic, legal, and transgenerational using the respondents' data. Each implication is briefly discussed below.

5.2.1. Economic Implications

As discussed earlier in the section 'African Worldview and Healing in African Neo-Pentecostalism', the hybridity of ANPLs and African traditional healing processes has both benefits and challenges. Despite the good and appreciable benefits of healing, the potential for extremism and the overdrive of healing and miracles activities can lead to the commercialisation of the process. It can also have economic implications for healing recipients. For example, selling handkerchiefs, holy water, honey, and anointing oil for outrageous prices or charging exorbitant fees for healing consultancy can lead to not only the abuse of the hybrid healing advantage but also the economic exploitation of healing recipients.

5.2.2. Social Implications

Earlier, this article noted the benefits of ANPLs' hybrid leadership vis-a-vis succession plans. However, this hybridity also has some challenges, one of which is social in nature. MRMUSHI, DRJOSHUA, DRJULIUS, and DRTHABISO attested to family division and communal disharmony following issues emanating from accountability, transparency, church ownership, and succession plans. MRMUSHI cited the example of a succession battle in the International Pentecostal Holiness Church (IPHC), where there was a family battle for leadership in the Shembe Church between Mduduzi and Vela Shembe, his uncle in Kwazulu Natal. DRJULIUS also cited the example of a Pentecostal church called the Christian Missionary Fellowship in Cameroon, where the founder's family has been divided because of a succession plan. In addition, MRMUSHI stated that in recent times, nothing has divided the church more than money or financial matters. In various communities, money has divided kings, chiefs, and families. Similarly, when the ANPLs are financially unaccountable or when the BOT members and related stakeholders disagree with the leadership style of the ANPLs on financial matters, it can lead to social tension and divisions among the Christian community.

5.2.3. Transgenerational Development Implications

This relates to the development of future leaders. Although some respondents agreed that syncretic gerontocracy is good for the church, others argued that it has the potential to limit the capacity and development of the younger generation of leaders. As PJOSHUA stated, "A big Cock does not allow a small Cock to crow," implying that giving preference to elders in all cases may have developmental implications. The older generation is likely to maintain traditions, whereas the younger generation may opt for change. The question then is whether the church wants to only maintain traditions or consider a hybrid of tradition and change. Since change through innovation can engender development, when innovative young ministers are not trained and offered opportunities for leadership, church underdevelopment and the proliferation of traditional practices can easily occur. Moreover, the younger generation is the future of the church. Any form of extremism in gerontocratic leadership can affect the successful development of future leaders.

5.2.4. Legal Implications

In a recent article, it was argued with empirical evidence that some Neo-Pentecostal leaders have abused adherents sexually, physically, economically, and emotionally (Orogun and Pillay 2022, pp. 1–28). Likewise, the respondents suggested that extremism in the spiritual authority or submissive theology of some ANPLs may engender abusive behaviour. FOPSA strongly believed that most projects and church businesses registered in the name of the General Overseer and their family members were financed through fundraising during

church services, with the initial manipulative message that these businesses belonged to the church and that all members were entitled to share the benefits. JOPSA seconded the allegation and claimed that it was a form of larceny and criminal behaviour. In his words, 'The schools were not started as private or personal schools, they are institutions built from church funds and the highest givers are the poor within the church whose children cannot afford such schools' (see Orogun 2020, pp. 177–80). Correspondingly, Falaye and Okanlawon alleged larceny and criminal conversion via the personalisation of church schools (see Okanlawon 2018, p. 37; Falaye 2017, pp. 3–4). Besides church-owned schools, there could be other areas where larceny and criminal conversion are prevalent. The testimony of CHUDAP was that some ANPLs registered their businesses under the church's name, thereby evading taxes. This practice contravenes tax laws and questions the legality of the ANPLs' registered businesses. DRJULIUS noted the government policies for NGOs in South Africa, where it is expected that church leaders will separate church ownership from the General Overseer's personal businesses or investments. Sadly, some ANPLs do not comply with these regulations. Thus, the adoption of the African monarchical principle of ownership by the ANPLs may conflict with existing laws.

### 5.3. Critique

The goal of this section is to assess African Neo-Pentecostal hybrid leadership through the lens of Jesus's leadership, focusing on the four variables: accountability, succession plans, gerontocracy, and healing.

### 5.3.1. Accountability and Transparency

In the previous section, it was established that hybrid leadership regarding accountability can have economic, legal, and social implications. Indeed, having authority without accountability can be disastrous. Wiersbe noted that many Christian leaders enjoy the authority and prestige of office and forget about the "tremendous responsibility and accountability" (Wiersbe 1982, p. 358). The hero worship and superhuman status, which confer unquestionable authority on the ANPLs, seem to conflict with Jesus's views. In Matthew 12:36, the clause of accountability excluded no one and stated that 'everyone will have to give account' (NIV). Thus, Jesus recognised accountability as one of the fundamental qualities of leadership. In addition, the feeding of five thousand people provides a good case study of accountable leadership, as demonstrated in Matthew 14:13–21, Mark 6:31–44, Luke 9:12–17, and John 6:1–14. In this scenario, the disciples accounted for the 12 baskets of food that were left over after everyone had been fed. This implies that the disciples may have learnt the idea of accountability by keeping records of budget surplus (12 Baskets) and not letting food go to waste under the leadership of Jesus. Thus, accountability may not be excluded from Jesus's view of responsible leadership. Similarly, regarding watchfulness, in Luke 12:48, Jesus stipulated that to whom responsibility is given, commensurate accountability will be required.

By inference, providing answers to questions from church adherents in all areas of church management may not be antithetical to Jesus's philosophy of accountable leadership. There is no scriptural provision that exonerates leaders from accountability, and likewise, there is no scriptural verse that contradicts adherents' demands for records. It is incontrovertible to infer that accountability is a responsibility and not a choice. With this background, ANPLs need to be cautious when blending missional accountability with African traditional systems of kings, elders, and related patriarchal ideas.

### 5.3.2. Ownership and Succession Planning

In the previous sections, the challenges of hybridity vis-a-vis ownership and succession planning were discussed extensively. All the associated legal, social, and economic challenges are driven by extremism. However, when there is a commitment to responsible leadership, the benefits of hybrid succession planning to ANPLs can be enormous. Some respondents' views validated the benefits of hybrid leadership in succession planning. They

strongly agreed that God's call was the most acceptable yardstick for selecting ministry successors and that if ANPLs' family members were qualified by this calling, there was nothing wrong with the idea. The involvement of Jesus's family in the New Testament provides a good case study. Bible scholars identified James, Joses, Simon, and Jude as the brothers of Jesus in Matthew 13:55 and Mark 6:3 (see Lovorn 2011). Three views have been advanced by scholars on the siblings of Jesus: that they are his actual siblings, his stepbrothers, or his cousins (Bibleinfo 2023; Lovorn 2011). In all these views, the underlying idea is that they are members of Jesus's family. This case study reveals that Jesus's family members were not strangers to his ministry. For example, Jesus's mother and brothers accompanied him to Capernaum after the marriage at Cana in John 2:12. In Acts 1:14 they were all present when the Holy Spirit came upon believers at Pentecost. Much later, Jesus appeared after his resurrection to his brother James, which led to his conversion in 1 Corinthians 15:7, and by the middle of the first Christian century, James had become the leader of the church in Jerusalem (Acts 15:13; Galatians 1:18–19 and 2:9). Paulson (2023) in his reference to James 1:1 called James the chief spokesman for the Jerusalem church in addition to the fact that he was the author of the book of James. Jude also wrote a book in the New Testament as documented in Jude 1:1. These examples reveal that in the ministry of Jesus on Earth, his family members were not excluded from leadership responsibility, just as the African tradition of succession planning prioritizes family members' participation. Therefore, the adoption of the African model of succession planning by the ANPLs is in line with Jesus's model of leadership. However, caution against extremism needs to be taken to avoid the abuse of privilege in leadership. The family of Jesus presents a good example, as there are no records to show that his brothers abused any privileges. James, in particular, was seen as a wise man of inestimable value to the church of Jerusalem.

### 5.3.3. Gerontocracy

A few respondents asserted that gerontocracy was advantageous. Conversely, others believed that it promoted the exclusion of the younger generation from leadership roles. Whether the respondents spoke for or against it, the key factor here is balance. The Bible in Lev 19:32, Job 12:12 and 1kings 12:6 recognises the role and the wisdom of the elders in the church. Interestingly, Jesus's brother, who heads the Jerusalem church, was called Elder James. However, Jesus's principle of inclusion created a space for the younger generation. The word "children" that is used in Mark 10:13–16 can pass for a description of the younger generation. Jesus forbade the disciples (elders personified) from hindering the younger generation. He also debunked thoughts that undervalued or underutilised the younger generation, as he asked the elders (disciples), "Who is the greatest?" and addressed their hierarchical mindset with his words, "Whoever takes the lowly position of this child is the greatest in the kingdom of heaven" (Matthew 18:1–5, NIV). Thus, in some cases, age may not necessarily be an advantage in kingdom service. In this sense, hybrid leadership via gerontocracy is valuable but it must not exclude the participation of the younger generation.

### 5.3.4. Healing and Miracles

It was argued earlier that hybrid leadership in the context of healing by the ANPLs was a welcome development. The use of elements such as water, salt, oil, and honey was easy because the ANPLs adopted the African worldview. Interestingly, this is consistent with Jesus's healing ministry. In John 9:6–7, Jesus used mud mixed with saliva and water to administer healing. In addition, in Luke 4:40 he performed healing through the laying of hands. James, the brother of Jesus who was assumed to understand the ministry of Jesus, called for the use of oil to anoint the sick alongside the prayer of faith (James 5:4). These three scenarios indicate that using elements such as soil or mud, saliva, honey, or water alongside prayers of faith in the name of Jesus is a biblical practice. With this background, the hybridisation of African healing processes, especially the use of herbal therapeutic procedures alongside prayers by the ANPLs, is appreciable. However, caution must be taken against extremism, especially the adoption of practices such as human sacrifice in

the name of healing. Just as Moreau observed, the ANPLs must avoid the "dilution of the essential gospel truths due to the incorporation of non-Christian elements" (Moreau 2000, p. 924).

## 6. Final Remarks

Since it has been established that hybrid leadership has both benefits and challenges that require caution, how can the ANPLs appropriate it with caution? The four variables representing the four areas of leadership in the context of this article require caution. This article suggests that being cautious means responsibly applying or appropriating hybrid leadership. The first point of caution is to set *boundaries* against extremism. This article submits that ANPLs cannot go wrong with creating boundaries. Consequently, legal implications such as larceny and criminal conversion can easily be avoided. In addition to setting boundaries, a *heart of service* is imperative for exercising caution. When leaders have a heart of service, service comes to the fore and self-centredness takes a back seat, making it difficult to assume the posture of extremism. In Matthew 20:20–28, Jesus presented the heart of service as a core value of responsible leadership. When hybrid leadership is approached from the perspective of servant leadership, the proclivity to commercialise healing or extort and exploit subscribers or adherents can be avoided. Likewise, taking the posture of a servant will naturally help the ANPLs to see the younger generation differently, not as those with less wisdom and capacity, but as those to be nurtured, supported, and empowered over time. The third point of caution is that the ANPLs should always *subject every hybridisation process to a moral question*. For example, when blending prayers with African traditional healing processes, the moral question could be a scale for evaluating actions such as human sacrifice. Jesus's context is also very important when responding to moral questions. For example, in Matthew 5:21, Jesus restated the Old Testament law of "Thou shall not kill." This instruction provides a scale for measuring the syncretic participation of ANPLs in any healing process that requires human sacrifice. Finally, in these three points of caution, the denominator is balance. Hybrid leadership can never go wrong with a balanced approach. The quest for balance can help to overcome the proclivity for extremism.

**Funding:** This research received no external funding.

**Institutional Review Board Statement:** The study was conducted in accordance with the Declaration of Helsinki, and approved by the Research Ethics Committee of Faculty of Theology, University of Pretoria (protocol code T013/22 and date of approval: 4 May 2023).

**Informed Consent Statement:** Informed consent was obtained from all subjects involved in the study.

**Data Availability Statement:** Not applicable.

**Acknowledgments:** The author thanks the Guest Editors for providing support for this article.

**Conflicts of Interest:** The author declares no conflict of interest.

## Notes

[1] Historically, Neo-Pentecostalism emerged in the 1950s and 1960s, but here, it refers to the newer African versions from the 1980s to date. These include the faith and miracle, prophetic, healing, and deliverance movements in Africa. Note that these newer African versions are categorised as classical, contemporary, and paradigm Neo-Pentecostal churches (Orogun 2020, pp. 10–11, 24–25).

[2] Submissive theology is a concept of religious manipulation, where religious leaders coax their followers into blind adherence, which rejects independent thought or decision making informed by intellectual reasoning. This concept emerged from the Pan-African philosophical consciencism of Fela Anikulapo, who suggested that Christianity and Islam were foreign religions used by their leaders (Arabs and Westerners) to suppress Africans' capacity to reason and fight against modern slavery. In his opinion, the two religions use the concept of Arabisation and Westernisation in collaboration with African associates through a subjugating theology to sustain modern slavery in Africa. He further claimed that these African associates included pastors

and imams who use their titles and positions to secure advantages to silence the masses in the face of colonial and post-colonial oppression (see Olaniyan 2009, pp. 88–89).

3    General Overseer is the self-conferred title by the founding or presiding leaders of African Neo-Pentecostal churches, especially in South Africa, Nigeria, and Ghana. It is to some degree a position of supremacy, authority, and command. In some cases, it leads to authoritarianism.

4    A system of initiating young people into adulthood in a traditional way.

5    Kabi-o-osi is the etymological root word for 'Kabiyesi' in the Yoruba language of western Nigeria. Kabiyesi is the name given to kings, whereas the root 'kabi-o-osi' is the functional definition of the name. Kabi-o-osi means 'you are unquestionable'. By implication, kings, leaders, fathers, and all men and women in positions of authority cannot be questioned or challenged. They are semi-divine beings whose actions are justified in their own rights.

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
