# Peer review of "Hybrid Leadership in African Neo-Pentecostalism"

_religions, doi:10.3390/rel14050632_

Round 1

Reviewer 1 Report

The author poses worthwhile questions about the leadership structures of Pentecostal churches in Africa. The focus on similarities between traditional African hierarchies and Pentecostal theologies is an interesting one. As the author rightly notes, these topics consistently take on new emphasis and urgency in new surroundings. This has been a hallmark of both early Pentecostalism and neo-Pentecostalism. The article’s main strength is its relevance to the ongoing scholarly discussions about how and why Pentecostalism grows in various contexts. However, the paper has several pressing issues that should be resolved prior to publication.

To begin, the article needs a clear discussion of its rationale for employing the term ‘syncretism’ as opposed to more recent adoptions of ‘hybridity.’ Scholarly debates about ‘syncretism’, ‘hybridity’, and ‘cultural additivity’ are almost completely ignored in the work. In fact, many of the sources cited by author are rather dated and do not reflect the current state of the field. The article would benefit from the inclusion of a discussion of the wider trends in scholarship around these terms if only to clarify why the author chose ‘syncretism’ as the most descriptive lens through which to approach the processes identified in the paper. The author might consider reviewing Ackermann’s research (2011) on cultural hybridity. 

Another area for improvement throughout the paper is the author’s use of broad terms without proper description including references to ‘submissive theology’ (p. 2, ln 85), ‘syncretism view’ (Table 1), and African Independent Churches (p. 8, ln 318). The author assumes readers, especially readers outside the field, understand these terms. The author should provide additional descriptions/definitions in the text or footnotes.

In line with this, the paper does not provide a clear description of Pentecostal belief, nor does it differentiate between the various strains of Pentecostalism. A reader familiar with the topic might assume the author mostly refers to neo-Pentecostal incarnations of the movement, but this is not assured.

Additionally, the author should consider providing some comparison, if briefly, between the structure of Pentecostal leadership in Africa and other regions of the world. Pentecostalism has set deep roots in other parts of the world. Including a comparison will further underscore the author’s point that APLs are syncretic mixtures of Pentecostal beliefs and traditional African worldviews. Pentecostalism, and especially its neo-Pentecostal forms, are widely known to be hierarchical and patriarchal. The reader is left asking – How are the APLs different?

The author may also want to rethink the critiques and conclusions listed below 6.3. This is not to suggest the points are not true or inaccurate. However, they read as prescriptive and subjective. Statements like “The Pentecostal culture where only the General Overseer determines the management of the church is socially, morally, and scripturally incorrect (p. 13, ln 566-68) and “…whatever healing practices are to be syncretized in the church must be scripturally evaluated before application.” (p. 15, ln 668-9) reflect the author’s biases and fall outside the scope of the paper’s intent. While the author may support these opinions, they are ideas that are not fully explored in the paper and have the unintended consequence of diminishing believers’ agency.

Finally, the article needs significant stylistic and grammatical revision. Several sentences are confusing and detract from the overall quality of the paper. Examples include (the list is not exhaustive):

p. 3 ln 105 – G.O. missing second period/’General Overseer’ not used until later in the article

p. 4 ln 106 – BOT never described until later in the article (Board of Trustees)

p. 4 ln 107 – Contrariwise capitalized?

Author Response

Hello, Please see the attachment.

Reviewer 2 Report

First of all, let me congratulate the author(s) of this research on such valuable contribution to the study of Pentecostalism worldwide. I have no doubts that this article meets the standards, and thereby deserves to be published, not least because of the issues it raises relative to African Pentecostalism and the data it offers to other researchers in this area. Yet, kindly, allow me to make two general remarks with respect to the theoretical framework (not to the qualitative research, which is fine).

The first remark concerns the coherence of the whole text and, more particularly, the very concept of syncretism, which seems to hold the whole text together. If it applies “that syncretic elements are inherent in all religions” (Hughes 1988, p.670 / line 36), why should it represent a problem in African Pentecostalism? The author(s) even gos on to conclude that, “[b]y implication, there is no [historical] period without syncretism (line 48)”. Therefore, if the author(s) is providing a framework that validates and even warrants syncretism, what would be the point of criticising it? If the article’s point of departure is that syncretism is a religious and/or cultural constant, the inquiry, in my opinion, ought to be formulated not along the lines of empirical evidence, but of, conceivably, theological justification. In other words, not by providing evidence of its existence, but by analyzing how these actors (i.e. APLs) undergird (biblically, theologically) the practices adopted in their churches (gerontocratic leadership, non-transparency, ownership, succession plan, healing and miracles). If, conversely, the aim is to criticize syncretic practices, the theoretical framework that the whole study is based upon is to be reconsidered.

My second remark relates to the juxtaposition made by the author(s) between the African Pentecostal Leaders model and the “Jesus’ servant leadership model”, whereby the latter is suggested as a normative biblical doctrine with which the former is contrasted, criticized, and delegitimized. In my opinion, the “Jesus’ servant leadership model” is not sufficiently elaborated on, in the first place. Even though one may be able to make a few associations with this term, the scriptural quotations provided fall short of substantiating it theologically. Furthermore, the biblical foundations provided are not followed by any theological reflection upon this concept. Second, since the APLs model is neither captured theologically/ scripturally (along the lines of, let us say, a Jesus’ King leadership model or anything else to that effect), any contrastation with it appears to be problematic. After all, it might as well be the case that APLs operate with the idea of Jesus servant, too, albeit with a different understanding of it. So, the question that arises to me is, theologically speaking, which model is the author(s) contrasting the “Jesus’ servant leadership model” with? This could, in my view, be spelled out more explicitly and/or in more detail.

In addition, please consider the following spelling mistakes (the number refers to the article’s line number, followed by the phrase in question):

200 However, Admittedly,

347 Moreso,

351 Beran Christianity as in Act 17.

500 Even in communities’

678 Moreso,

Finally, please consider the lack of consistency while employing the possessive ‘s’ with the name Jesus. At times, the author(s) uses Jesus’, at other times Jesus’s.

Round 2

Reviewer 1 Report

I applaud the author's thorough revision of the original manuscript. The changes adequately address the suggestions outlined in my original review.